# Regional variation in health is predominantly driven by lifestyle rather than genetics

Carmen Amador [1], Charley Xia[1], Réka Nagy[1], Archie Campbell [2,3], David Porteous[2,3], Blair H. Smith [3,4], Nick Hastie[1], Veronique Vitart[1], Caroline Hayward [1], Pau Navarro [1] & Chris S. Haley[1,5]

Regional differences in health-related phenotypes have been detected between and within countries. In Scotland, regions differ for a variety of health-related traits and display differences in mean lifespan of up to 7.5 years. Both genetics and lifestyle differences are potential causes of this variation. Using data on obesity-related traits of ~11,000 Scottish individuals with genome-wide genetic information and records of lifestyle and socioeconomic factors, we explored causes of regional variation by using models that incorporate genetic and environmental information jointly. We found that variation between individuals within regions showed substantial influence of both genetic variation and family environment. Regional variation for most obesity traits was associated with lifestyle and socioeconomic variables, such as smoking, diet and deprivation which are potentially modifiable. There was limited evidence that regional differences were of genetic origin. This has important implications for healthcare policies, suggesting that inequalities can be tackled with appropriate social and economic interventions.

[1] MRC Human Genetics Unit, Institute of Genetic and Molecular Medicine, University of Edinburgh, Edinburgh EH4 2XU, UK. [2] Centre for Genomic and Experimental Medicine, Institute of Genetic and Molecular Medicine, University of Edinburgh, Edinburgh EH4 2XU, UK. [3] Generation Scotland, Centre for Genomic and Experimental Medicine, Institute of Genetic and Experimental Medicine, University of Edinburgh, Edinburgh EH4 2XU, UK. [4] Division of Population Health Sciences, University of Dundee, Dundee DD2 4RB, UK. [5] Roslin Institute and Royal (Dick) School of Veterinary Studies, University of Edinburgh, Edinburgh EH25 9RG, UK. Pau Navarro and Chris S. Haley contributed equally to this work. Correspondence and requests for materials should be addressed to C.S.H. (email: Chris.Haley@roslin.ed.ac.uk)

Marked differences in health-related traits and diseases exist between countries and between regions within countries[1–4]. Regions in Scotland differ for several health-related traits, and display differences in lifespan of up to 7.5 years in men and 4.9 years in women[5]. Complex traits related to obesity, such as body size, have a genetic basis with heritabilities between 30 and 60%[6, 7]. On the other hand, changes in body size and the so-called global obesity pandemic are usually associated with environmental changes related to diet, exercise levels and other socioeconomic changes[8, 9]. From this perspective both genetics and lifestyle differences are potential causal factors for the observed regional differences in health-related traits[10–15]. Disentangling the underlying causes of variation in health-related traits has direct implications for the welfare of future generations; however, genetic and environmental variation may be closely associated and therefore difficult to separate.

In this study, we explored causes of regional variation in data on obesity-related traits as indicators of the health status of ~11,000 Scottish individuals with genotypic records and a variety of measurements of possible causal lifestyle and socioeconomic factors. Our aim was to exploit these high-quality genomic data and high fidelity and deep phenotypic, lifestyle and socioeconomic data to identify the factors contributing to the differences between regions in health-related traits. As expected, obesity-related traits such as body mass index and weight differ significantly between regions. By accounting for both the genomic data and the environmental information in our analyses, we showed that trait variation is substantially influenced by both genetic variation and family environment. However, the regional variation for most obesity traits was associated with lifestyle and socioeconomic variables, such as deprivation, physical activity, etc. rather than the regional genetic structure of the sample. These results imply that although genes and family environment are important determinants of health-related traits, regional differences are attributable mostly to potentially modifiable environmental factors.

## Results

**Overview of analyses.** The objective of this work was to disentangle genetic and environmental components of health-related traits, linked to geographic variation. We explored a Scottish population consisting of ~11,000 individuals with different degrees of kinship, genotyped for ~500K markers, phenotypes for 11 traits (8 anthropometric and 3 metabolic traits), geographic covariates (principal components) reflecting the regional genetic structure of the data (gPCs) and a large set of environmental covariates (socioeconomic and lifestyle (SELS)). We fitted jointly genetic and environmental information in a range of statistical models, in an innovative approach to disentangle the causes of regional variation. For more information, see Supplementary Tables 1, 2 and 3 and Supplementary Methods. An overview of analyses and models is shown in Fig. 1.

**Regional differences in traits within Scotland.** In order to illustrate the geographic differences existing in Scotland, in the Basal Model (Fig. 1) we adjusted each trait for sex, age and clinic and tested the traits for differences between the 32 regions (council areas, defined from the individual's postcode of residence; Supplementary Table 3). For 9 of the 11 traits studied, differences between regions (i.e., council areas) were significant at a 0.05 level (see Table 1, first column).

To test if the regional differences detected were due to the genetic relatedness of the sample, we adjusted for kinship by fitting a genomic relationship matrix (**G**) together with sex, age and clinic in a mixed model analysis (Family Model). We tested the residuals from this model for remaining regional differences. When including the genomic relationship matrix in the model, the differences between regions disappeared for two traits (height and body fat measured by bioelectric impedance analysis (BIA fat) (Table 1, second column), suggesting that the regional variation detected in the Basal Model for these two traits was due to the genetic relatedness of the sample. Nonetheless, for waist circumference, hips circumference, waist-to-hips ratio (WHR),

| Model | | Adjustment | Regional differences | Tested cause |
|---|---|---|---|---|
| Basal (B) | – | Trait ~ sex + age + clinic | Are there significant differences between regions? | Differences in the sample |
| Family (F) | Benchmark | Trait ~ **G** + sex + age + clinic | | Differences due to family structure |
| | Full | Trait ~ **G** + **K** + **C** + **S** + sex + age + clinic | | |
| Structure (S) | Benchmark | Trait ~ **G** + gPCs + sex + age + clinic | | Differences due to regional genetic structure |
| | Full | Trait ~ **G** + **K** + **C** + **S** + gPCs + sex + age + clinic | | |
| Environment (E) | Benchmark | Trait ~ **G** + SELS + sex + age + clinic | | Differences due to the recorded environmental factors |
| | Full | Trait ~ **G** + **K** + **C** + **S** + SELS + sex + age + clinic | | |
| Structure & Environment (S+E) | Benchmark | Trait ~ **G** + gPCs + SELS + sex + age + clinic | | Effect of environmental factors and genetic structure together |
| | Full | Trait ~ **G** + **K** + **C** + **S** + gPCs + SELS + sex + age + clinic | | |

**Fig. 1** Overview of the models and analyses performed. G, Genomic Relationship matrix; K, Kinship matrix; C, Couples matrix; S, Siblings matrix; gPCs, Geographic Principal Components; SELS, Socioeconomic and lifestyle covariates

**Table 1 Significance of region on phenotypes in the benchmark framework**

| Trait | Model | | | | |
|---|---|---|---|---|---|
| | Basal | Family | Structure | Environment | Structure and environment |
| Height | 3.50E-06* | 0.433 | 0.573 | 0.939 | 0.957 |
| Weight | 0.129 | 0.410 | 0.413 | 0.811 | 0.805 |
| BIA fat | 0.002* | 0.100 | 0.141 | 0.466 | 0.464 |
| Waist | 1.10E-05* | 0.009* | 0.016* | 0.185 | 0.195 |
| Hips | 0.070 | 0.293 | 0.302 | 0.593 | 0.588 |
| WHR | 4.52E-07* | 0.001* | 0.004* | 0.157 | 0.196 |
| BMI | 9.46E-06* | 0.006* | 0.011* | 0.387 | 0.406 |
| ABSI | 2.70E-04* | 0.003* | 0.005* | 0.016* | 0.015* |
| Creatinine | 1.81E-11* | 2.09E-06* | 3.06E-06* | 4.29E-06* | 5.81E-06* |
| TC | 0.250 | 0.374 | 0.372 | 0.464 | 0.461 |
| HDL | 1.98E-05* | 0.008* | 0.010* | 0.184 | 0.188 |

The values show the significance ($p$-values) of region in four models: 1. differences in the traits (first column, Basal Model); 2. adjusting for kinship (second column, Family Model); 3. adjusting for kinship and genetic structure (third column, Structure Model); 4. adjusting for kinship and environmental covariates (fourth column, Environment Model); 5. adjusting for kinship and genetic and environmental covariates (fifth column, Structure and Environment Model). An asterisk marks the estimates that are significantly different from zero. All models adjusted for sex, age and clinic

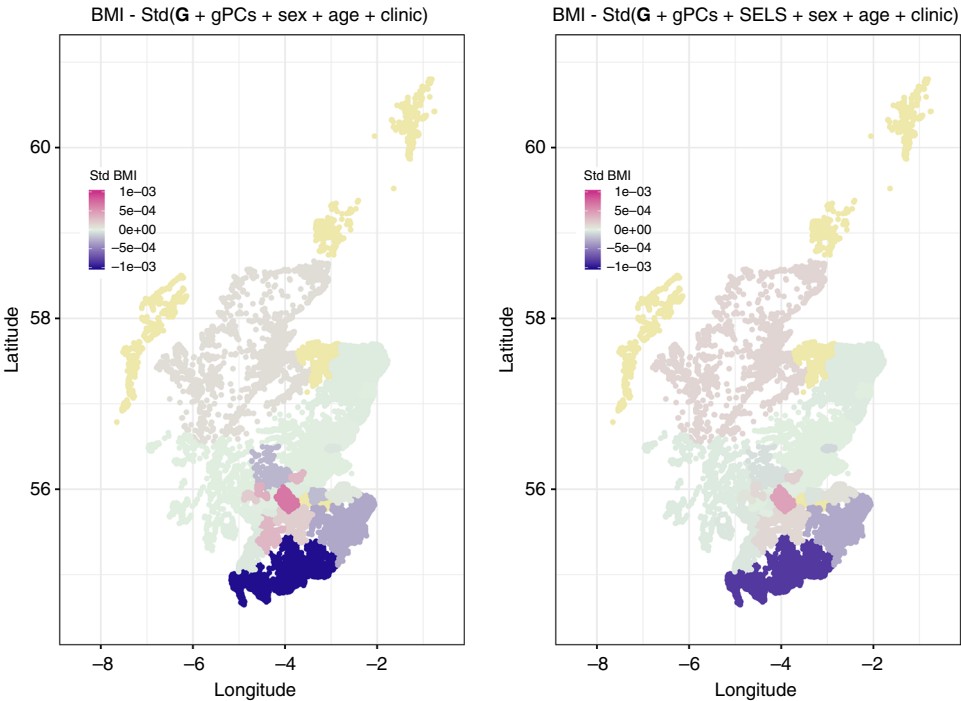

**Fig. 2** Regional values of BMI before and after adjusting for the environmental variables. Changes in the standardised means of BMI per region before (*left panel*) and after (*right panel*) adjusting for all the lifestyle and socioeconomic covariates. *Yellow*: regions with less than 20 individuals, not considered

body mass index (BMI), a body mass index (ABSI), creatinine and high density lipoprotein (HDL) levels regional differences still exist ($\alpha = 0.05$) after adjusting for the genetic relatedness and family structure in the sample.

We then explored if the regional differences could be explained by the population genetic structure of the sample, i.e., the genetic differences between the regions. To do that we adjusted for ten geographic principal components (gPCs) that represent geographical population genetic structure in the cohort. The gPCs were calculated using a subset of unrelated individuals and unlinked markers and then extrapolated to the rest of the population. They reflect the genetic differences between regions as shown in Amador et al.[14] (for more information see Methods). We adjusted for the gPCs (together with a genomic relationship matrix, sex, age and clinic in the Structure Model) and we used the residuals of the model to test whether the

regional differences remained significant (Table 1, third column). For all six traits with significant regional differences after the previous analyses, these differences remained significant ($\alpha = 0.05$) after adjusting for the gPCs, i.e., the genetic differences between regions do not explain the regional differences in the studied traits.

Next, we examined if the regional differences could be explained by the environmental differences measured in the cohort by adjusting for the SELS covariates. We fitted a model adjusting for a genomic relationship matrix and SELS covariates, representing this environmental information, together with sex, age and clinic (Environment Model). When we tested the significance of the region in the residuals of this model (Table 1, fourth column), we observed that only ABSI and creatinine showed significant differences ($\alpha = 0.05$) between regions and these differences had become non-significant for waist

| Table 2 Proportion of the phenotypic variance explained by genomic (G: genomic relationship matrix, K: kinship matrix) and environmental matrices (C: couple matrix, S: sibling matrix) | | | | | |
|---|---|---|---|---|---|
| **Trait** | **Model** | **G** | **K** | **C** | **S** |
| Height | Model F | 0.481 (0.034)* | 0.387 (0.043)* | 0.125 (0.032)* | 0.007 (0.018) |
| | Model S+E | 0.457 (0.034)* | 0.415 (0.042)* | 0.122 (0.032)* | 0.006 (0.017) |
| Weight | Model F | 0.274 (0.034)* | 0.349 (0.046)* | 0.200 (0.035)* | 0.030 (0.021) |
| | Model S+E | 0.280 (0.034)* | 0.333 (0.046)* | 0.191 (0.035)* | 0.027 (0.021) |
| BIA fat | Model F | 0.274 (0.034)* | 0.229 (0.047)* | 0.207 (0.036)* | 0.049 (0.023)* |
| | Model S+E | 0.248 (0.035)* | 0.212 (0.048)* | 0.179 (0.037)* | 0.052 (0.023)* |
| Waist | Model F | 0.198 (0.033)* | 0.341 (0.047)* | 0.241 (0.034)* | 0.029 (0.022) |
| | Model S+E | 0.182 (0.034)* | 0.320 (0.047)* | 0.220 (0.035)* | 0.029 (0.023) |
| Hips | Model F | 0.192 (0.034)* | 0.314 (0.047)* | 0.211 (0.036)* | 0.043 (0.023) |
| | Model S+E | 0.197 (0.034)* | 0.297 (0.047)* | 0.200 (0.036)* | 0.039 (0.023) |
| WHR | Model F | 0.152 (0.033)* | 0.205 (0.046)* | 0.120 (0.038)* | 0.022 (0.024) |
| | Model S+E | 0.099 (0.034)* | 0.217 (0.047)* | 0.089 (0.038)* | 0.022 (0.025) |
| BMI | Model F | 0.276 (0.033)* | 0.327 (0.046)* | 0.255 (0.034)* | 0.024 (0.021) |
| | Model S+E | 0.266 (0.034)* | 0.294 (0.046)* | 0.232 (0.035)* | 0.026 (0.022) |
| ABSI | Model F | 0.101 (0.032)* | 0.223(0.045)* | 0.051(0.037) | 0.025(0.024) |
| | Model S+E | 0.080(0.033)* | 0.245 (0.046)* | 0.049(0.037) | 0.021(0.024) |
| Creatinine | Model F | 0.247(0.035)* | 0.387 (0.046)* | 0.141 (0.032)* | 0.040 (0.022) |
| | Model S+E | 0.223 (0.035)* | 0.393 (0.047)* | 0.138 (0.032)* | 0.050 (0.022)* |
| TC | Model F | 0.205 (0.035)* | 0.143 (0.048)* | 0.045 (0.035) | 0.081 (0.024)* |
| | Model S+E | 0.208 (0.035)* | 0.131 (0.048)* | 0.038 (0.036) | 0.086 (0.024)* |
| HDL | Model F | 0.311 (0.035)* | 0.267 (0.046)* | 0.093 (0.034)* | 0.038 (0.022) |
| | Model S+E | 0.299 (0.035)* | 0.238 (0.047)* | 0.060 (0.035) | 0.050 (0.022)* |

Columns three to six show the variance captured by each matrix when fitted together (Full Model). The two models used were the Family Model (F): with sex, age and clinic included as covariates; and the Structure and Environment Model (S+E): with gPCs, SELS, sex, age and clinic included as covariates. An asterisk marks the estimates that are significantly different from zero

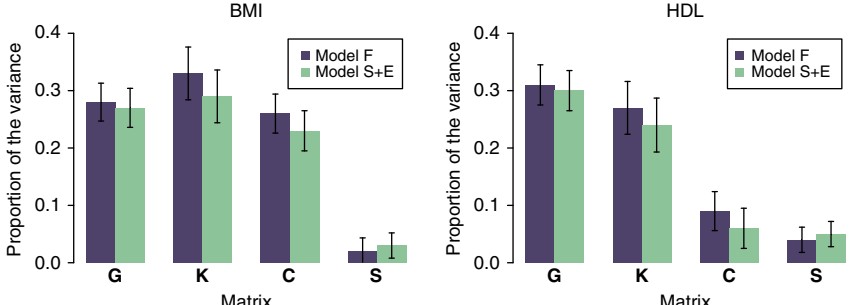

**Fig. 3** Heritability estimates from models with different covariates. Proportion of the variance in two different traits captured by each of the genetic or environmental matrices fitted: Model F: including four matrices and sex, age, clinic as covariates (*blue bars*); Model S+E: including four matrices, gPCs, SELS and sex, age, clinic as covariates (*green bars*). *Error bars* show the standard errors of the estimates

circumference, BMI, WHR and HDL, indicating that the regional differences are explained by the measured SELS variables. We fitted a final model including both the gPCs and the SELS covariates (Table 1, Structure and Environment Model) to corroborate the results. The results obtained for this model were very similar to those from the Environment Model, reinforcing the conclusion that the SELS covariates are responsible for the regional differences observed.

A visualisation of the changes in the standardised residual means for each trait per region before and after adjusting for the SELS variables was created using latitude and longitude of Scottish postcodes in R[16]. This is shown for BMI in Fig. 2 and for all traits in Supplementary Fig. 1. The only remaining regional differences were for creatinine and ABSI. Since our results suggest that those were not due to the geographical population genetic structure (Table 1, Structure Model), these remaining differences are likely to be caused by other environmental variables not measured in our data and not associated with family genetic structure or family environment.

We repeated the whole set of analyses including a larger set of genetic and environmental matrices (**G**: genomic relationship matrix, **K**: kinship matrix, **C**: couples matrix, **S**: siblings matrix; see Fig. 1: Full models F, S, E and S+E) combining the different set of covariates and the results observed were similar to those of the Benchmark models described above: most regional differences were removed when fitting the SELS variables (Supplementary Table 4).

**Heritability estimates and covariate effects.** We evaluated the proportion of the variance explained by all the components fitted in several mixed models to further explore genetic and environmental variation in the 11 traits studied following Xia et al.[17] Using mixed-model analysis[18, 19] we partitioned the phenotypic variance into components representing genetic or environmental effects. We used two genetic relationship matrices (**G** and **K**) to account simultaneously for the genetic sharing among distant and closely related individuals[7]; and two environmental relationship

**Table 3 Proportion of the phenotypic variance explained by the covariates in the Benchmark Structure + Environment (S+E) Model**

| Trait | Socioeconomic | Lifestyle | gPCs |
|-------|---------------|-----------|------|
| Height | 0.0045 | 0.0019 | 0.0048 |
| Weight | 0.0090 | 0.1244 | 0.0000 |
| BIA fat | 0.0069 | 0.1020 | 0.0018 |
| Waist | 0.0134 | 0.0061 | 0.0022 |
| Hips | 0.0094 | 0.0211 | 0.0000 |
| WHR | 0.0069 | 0.0181 | 0.0037 |
| BMI | 0.0184 | 0.1400 | 0.0015 |
| ABSI | 0.0031 | 0.3526 | 0.0023 |
| Creatinine | 0.0001 | 0.1906 | 0.0004 |
| TC | 0.0040 | 0.0279 | 0.0000 |
| HDL | 0.0049 | 0.2830 | 0.0012 |

Socioeconomic comprises the covariates SIMD, years of education, household size, vehicle ratio and job status; lifestyle comprises the covariates alcohol units, smoking status, activity level and consumption of fruit, vegetables, fish, meat, eggs and dairy; gPCs are the 10 principal components of the genomic relationship matrix that describe the geographical population genetic structure

matrices that represented shared environments between members of a couple (**C**) and siblings (**S**)[17] (Fig. 1, Full models).

The proportion of the phenotypic variance explained by the components in a Full Model is shown in Table 2. The table includes the results for two types of analyses: the Family Model including only the matrices, sex, age and clinic, or the Structure and Environment Model (S+E) including the matrices and gPCs and SELS covariates together with sex, age and clinic.

The estimates of the genotyped-single-nucleotide polymorphism (SNP) heritability ($h^2_g$, proportion of the phenotypic variance captured by matrix **G**) and of the pedigree heritability ($h^2_{kin}$, captured by matrix **K**) did not change significantly when including the extended set of covariates in the model, even for those traits where the environmental covariates contributed to regional differences. Furthermore, for most of the traits the estimates of variance due to the shared environments of couples and siblings (**C** and **S**) were robust to the inclusion of the extended set of SELS variables (Table 2). This is illustrated for two traits in Fig. 3.

The proportion of the variance captured from the couple environment (matrix **C**) was significant for eight traits although for HDL the significance disappeared after including the full set of environmental covariates. This would suggest that, for HDL, some of the phenotypic similarities observed in couples can be accounted for by the recorded lifestyle or socioeconomic variables. In addition, the variance captured by the sibling environment (matrix **S**) was detectable only for two traits (BIA fat and TC). For creatinine and HDL, the variance captured by sibling environment was not different from zero in the Family Model, but became significant after including the whole set of covariates. In all the cases the differences in proportion of the variance captured between the Structure and Environment Model (including the whole set of covariates) and the Family Model explored were subtle.

Table 3 shows the variance explained by the SELS covariates together with the gPCs in the models including a **G** matrix (details of each individual covariate are shown in Supplementary Table 5). The amount of variance explained by SELS covariates ranged between 0.64 and 35.57% while the gPCs explained always < 0.5% of the variance for all traits. Scottish index of multiple deprivation (SIMD) was the covariate affecting most traits (all except for creatinine) and years of education also explained substantial variance for several traits, with effects on most of the body measurements. Activity level explained a large amount of

variance (up to 18.9%) for traits like HDL, BMI, weight and BIA fat. The dietary variables showed effects on many traits but overall explained little variance. For all traits the SELS covariates explained more variance than the geographical population genetic structure, which is consistent with the results showing that the regional differences in the obesity-related traits are associated with environmental rather than genetic variation between the regions.

## Discussion

Geographic differences in health-related phenotypes and diseases have been detected between countries and between regions within countries, and both genetics and environment could potentially account for these differences[11–14]. In this study, we disentangled the underlying causes of phenotypic differences between regions for 11 health-related traits in Scotland. To do so, we accounted for genetic structure together with environmental differences captured by environmental covariates and similarity matrices. We included all of them together in different mixed linear models in an innovative approach to the study of regional differences in health-related phenotypes. We showed the impact of familiar genetic structure, geographical population genetic structure and lifestyle and socioeconomic variables in all the traits. We found that for most of the obesity-related traits, existing regional differences within Scotland cannot be explained by geographical population genetic structure and they are predominantly driven by lifestyle and socioeconomic causes.

We showed that for height and BIA fat, the regional differences were explained by the genetic relatedness of the sample, disappearing when we corrected using a genomic relationship matrix. In the case of height, the geographical population genetic structure (gPCs) still explains 0.5% of the variance in the trait (Table 3 and Supplementary Table 5). In a previous study at the pan-European level, Robinson et al.[10] detected regional differences in the genetics of height. Our results suggest that the differences observed in our Scottish cohort are due to the genetic similarity and the relatively high variance explained by the gPCs concur with results reported in Robinson et al.[10] particularly considering the relatively small geographic range in our sample.

For most of the obesity traits, the regional differences disappeared when adjusting for a large set of measured environmental covariates, indicating that after accounting for other factors, the residual regional discrepancies were caused by environmental differences between the regions. An important implication is that studies that explore the causes of regional inequalities should account for both genetic and environmental factors if they are to avoid reaching biased conclusions, particularly in the presence of relatives. In the case of BIA fat, the results show a different pattern from the rest of the obesity traits. The gPCs explain a 0.18% of the variance, similar to BMI or waist circumference, but fitting the genetic relatedness of the sample explained regional differences. The phenotypic correlation between BIA fat and BMI is 0.62, suggesting that only a proportion of 0.38 of the variance was shared between traits.

The addition of the extended set of covariates did not change substantially the heritability or the variances captured by environmental matrices, suggesting that for this type of analysis, fitting only the basal covariates should suffice to obtain accurate heritability estimates for these health-related traits. Hence, previous studies that have not included these effects are unlikely to have produced significantly biased heritability estimates.

The deprivation index SIMD was the environmental variable significantly affecting the largest number of traits, although the variance accounted for by SIMD was relatively small for all traits (Supplementary Table 5). SIMD combines several indicators of

deprivation (such as level of income, education, employment, crime, etc. ref. [20]) in one index and the associations with SIMD corroborate the described associations between socioeconomic inequalities and health[11, 12]. The results showed that people living in less deprived areas are taller, with higher levels of HDL, and lower BMI and BIA fat levels. Other environmental variables also explained part of the differences between regions such as years of education, level of activity or alcohol intake. These more specific variables could be also picking up effects of more complex environmental variables such as socioeconomic status (e.g., if there is stratification in levels of alcohol intake or diet composition between different socioeconomic groups).

The increased prevalence of obesity is a worldwide health concern. Reducing the incidence of obesity by effective intervention policies in affected areas would provide substantial benefits in health and quality of life of individuals concerned and significantly reduce associated healthcare costs to the community[1, 4, 9]. In this study, we have shown that for most of the examined obesity traits, regional differences exist even after accounting for genetics and they can be explained by environmental differences between those regions. These environmental factors are potentially modifiable and therefore could be actionable from a health policy point of view, with the potential for appropriate interventions reducing inequalities in health between areas. In particular, our results show a large effect of physical activity and dietary choices for the studied traits. These two are recognised as critical behaviours affecting obesity and are usually targeted by policy makers[21]. Designing the interventions to improve these habits is important, but focusing in the relevant geographical (deprived) areas and making an impact on the relevant strata of the population will be crucial.

## Methods

**Data set.** We used the data from the Generation Scotland: Scottish Family Health Study (GS:SFHS)[22]. Ethical approval for the study was given by the NHS Tayside committee on research ethics (ref: 05/s1401/89). Governance of the study, including public engagement, protocol development and access arrangements, was overseen by an independent advisory board, established by the Scottish government. Research participants gave consent to allow both academic and commercial research.

Individuals were genotyped with the Illumina HumanOmniExpressExome-8 v1.0 or v1.2. We used PLINK version 1.9b2c[23] to exclude single-nucleotide polymorphisms (SNPs) that had a missingness > 2% and a Hardy–Weinberg Equilibrium test $P < 10^{-6}$. Markers with a minor allele frequency smaller than 0.05 were discarded. Duplicate samples, individuals with gender discrepancies and those with more than 5% missing genotypes were also removed. The resulting data set was merged with the 1092 individuals of the 1000 Genomes population[24] and a principal component analysis (PCA) was performed using GCTA[18]. Individuals more than ix standard deviations away from the mean of principal component 1 and principal component 2 were removed as potentially having African/Asian ancestry as shown in Amador et al.[14] After quality control individuals had genotypes for 519,819 common SNP spread over the 22 autosomes. Of the ~24,000 individuals in GS:SFHS, the number of individuals without missing values for any of the covariates used in our study was N = 11,118 (4646 males and 6472 females) so we used this set of samples for all the analyses in order to allow comparisons between the models.

**Phenotypes.** We used measured phenotypes for 11 complex traits classified as anthropometric (height, weight, body fat measured by bioelectrical impedance analysis (BIA fat), waist circumference, hips circumference, WHR(waist/hips); BMI (weight/height$^2$), ABSI[25] (Waist/(BMI$^{2/3}$×Height$^{1/2}$))) and metabolic traits (levels of creatinine, total cholesterol and HDL in serum. Natural logarithm transformations were performed for all traits except for height and BIA fat, to obtain approximate normal distributions. Phenotypes with values greater or smaller than the mean ± 4 standard deviations (after transformation and adjusting for sex, age and age$^2$) were set to missing (Supplementary Table 1). Boxplots for each trait of individuals living in each region (corresponding to different council areas) are plotted in Supplementary Fig. 2.

**Covariates.** We explored a large set of covariates representing potential environmental factors influencing differences between individuals in the study. We fitted these factors as putative predictors of trait variation in statistical models

as discrete or continuous covariates depending on their nature. The covariates lay in three categories: basic, socioeconomic and lifestyle. Basic covariates were sex, age, and clinic where the phenotypes were measured; socioeconomic covariates were SIMD (a deprivation ranking based on living area[20]), years of education, household size, vehicle ratio and job status. Lifestyle covariates are alcohol units consumption, smoking status, activity level, fruit units eaten per day and consumption of different foods (fruit, vegetables, fish, meat, eggs and dairy). A detailed description of these variables is shown in Supplementary Table 2 and additional information on how the quality control was performed is given in Supplementary Note 1.

Information on the postcode at which individuals were living at the time when their data were recorded was also available. The individuals were allocated to their corresponding council area based on these postcodes. A more detailed description on the correspondence between postcodes and regions is shown in Supplementary Table 3. The distribution or incidence of the covariates in the different council areas is plotted in Supplementary Fig. 3.

We also calculated a set of variables that represent genomic geographic origin through a PCA. To do so, we created a pruned subset of SNPs in approximate linkage equilibrium with each other and we removed markers from chromosome 6 in the major histocompatibility complex region and markers in the 8p23.1 region[14]. We kept only unrelated individuals (i.e., by removing one individual in each pair with a genomic relationship coefficient larger than 0.025). We performed a PCA in this subset ($N_{ind}$ = 7370, $N_{SNP}$ = 91,390), we calculated the loadings of the SNPs contributing to each of the first 10 principal components, and we computed the values for these principal components for the whole 11,118 individuals used in subsequent analyses. The resulting set of variables represents well the regional genetic structure of the sample as shown in Amador et al.[14] We refer to these 10 PCs as gPCs or geographical population genetic structure.

**Matrices.** We used design matrices representing genomic or environmental relationships as in Xia et al.[17]: **G** is a genomic relationship matrix (GRM) containing relatedness between pairs of individuals based on identity-by-state at the genotyped SNPs[19, 26]. **K** is a matrix representing pedigree relationships as in Zaitlen et al.[7] This is a modification of **G** obtained by setting all entries in **G** lower than 0.025 to 0. **C** is a matrix representing common environmental effects shared between couples. The matrix contains a value of 1 between pairs of individuals identified as members of a couple as in ref. [17]; **S** is a matrix representing common environmental effects shared between siblings. The matrix contains a value of 1 between pairs of individuals identified as siblings as in ref. [17]. **G** and **K** were calculated using GCTA[18]; The environmental matrices (**C** and **S**) were created using R version 3.1.1[16].

**Analyses.** A summary with names of models and analyses undertaken is shown in Fig. 1.

Firstly, to illustrate which traits show variation between the different regions in Scotland we explored in a simple linear regression if there were differences in the traits between council areas (Basal Model, B). Using the statistical package R[16], we testing the significance of the variable "region where individuals live" (region) in a linear model.

To explore if trait variation was due to genetic structure or to the environmental differences, we used variance component analyses. This way the models take appropriately into account the kinship in the sample. All the analyses were implemented in GCTA[18]. The basic general mixed linear model explored is shown in (1).

$$y = X\beta + g_g + \varepsilon,\qquad(1)$$

where $y$ is an $n \times 1$ vector of observed phenotypes with $n$ being the number of individuals, $\beta$ is a vector of fixed effects and $X$ is its design matrix, $g_g$ is an $n \times 1$ vector of the total additive genetic effects of the individuals captured by genotyped SNPs with $g_g \sim N(0, \mathbf{G}\sigma^2_g)$. $\varepsilon$ is an $n \times 1$ vector for the residuals.

We fitted different sets of covariates in this general framework to explore four different models:

The Family Model (F) included only sex, age and clinic as fixed effects.

The Structure Model (S) included the geographical principal components (gPCs), together with sex, age and clinic as fixed effects.

The Environment Model (E) included the SELS covariates, together with sex, age and clinic in the vector of fixed effects.

The Structure and Environment Model (S+E) included all gPCs and SELS, together with sex, age and clinic in the vector of fixed effects.

For each model, we predicted by the BLUP (best linear unbiased prediction) method the total genetic (and environmental if appropriate) effect of each individual together with their residuals. We tested if the residuals were significantly different between the regions to show if the differences between regions were explained by the models or remained unexplained. We also estimated the variance explained by matrix **G** ($\sigma^2_g$) in all these models and the variance explained by covariates included was calculated as

$$Var(Covariate) * b^2 / Var(Trait),\qquad(2)$$

where $b$ is the effect of the covariate estimated from the GREML analyses.

Model (S) including a genetic matrix (**G**) and the gPCs allowed us to test if adding the geographic structure to Model (F) (including only a genetic matrix) would account for the differences between regions. Model (E) including G and the SELS covariates compared with Model (F), allowed us to test if the differences were due to the SELS variables. Model (S+E) included both gPCs and SELS together for comparison with the two previous models.

We fitted again the models described above, including in addition another genetic and two environmental matrices fitted in linear mixed models as in Xia et al.[17] to test if the observed regional differences were affected by the inclusion or exclusion of the environmental matrices (**C**, **S**) and **K** as shown in (3)

$$y = X\beta + g_g + g_{kin} + e_c + e_s + \varepsilon, \qquad (3)$$

where $g_{kin}$ is an $n \times 1$ vector of the extra genetic effects associated with the pedigree for relatives with $g_{kin} \sim N(0, \mathbf{K}\sigma^2_k)$, $e_c$ and $e_s$ are $n \times 1$ vectors representing the common environmental effects shared by couples or siblings, with $e_c \sim N(0, \mathbf{C}\sigma^2_c)$ and $e_s \sim N(0, \mathbf{C}\sigma^2_s)$.

Results for the four models (Model (F), Model (G), Model (E) and Model (G+E)) were also explored in the context of Eq. (3).

**Data availability**. Data are available from the MRC IGMM Institutional Data Access/Ethics Committee for researchers who meet the criteria for access to confidential data. GS:SFHS data are available to researchers on application to the Generation Scotland Access Committee (access@generationscotland.org). The managed access process ensures that approval is granted only to research which comes under the terms of participant consent which does not allow making participant information publicly available.

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

## Acknowledgements

We acknowledge the Medical Research Council (MRC) UK for funding (grants MC_PC_U127592696 and MC_PC_U127561128). Generation Scotland has received core funding from the Chief Scientist Office of the Scottish Government Health Directorates CZD/16/6 and the Scottish Funding Council HR03006. Genotyping of the GS:SFHS samples was carried out by the Genetics Core Laboratory at the Wellcome Trust Clinical Research Facility, Edinburgh, Scotland and was funded by the UK MRC and the Wellcome Trust (Wellcome Trust Strategic Award 'STratifying Resilience and Depression Longitudinally' (STRADL) Reference 104036/Z/14/Z). We are grateful to all the families who took part, the general practitioners and the Scottish School of Primary Care for their help in recruiting them, and the whole Generation Scotland team, which includes interviewers, computer and laboratory technicians, clerical workers, research scientists, volunteers, managers, receptionists, healthcare assistants and nurses.

## Author contributions

C.S.H., P.N. and C.A. conceived and designed the experiments presented in this manuscript. D.P. and B.H.S. contributed to conceive and design the study population and phenotypic recording. D.P., B.H.S. and N.H. contributed to oversight of the study and sample collection. C.A. conducted the analyses. C.X. created the common environment matrices. R.N., A.C., C.H. and V.V. managed and maintained the data and performed the quality control. C.A., P.N. and C.S.H. wrote the paper. All authors discussed results, read and approved the final manuscript.

## Additional information

**Competing interests:** The authors declare no competing financial interests.

