## [Peer Review File · Nature Communications]

Reviewers' comments:

Reviewer #1 (Remarks to the Author):

Reviewer Critique:

In their study, entitled "Regional variation in health is driven by lifestyle, not genetics", Dr. Amador and colleagues examined differences in health-related traits and diseases between countries and between regions within countries, focusing on Scotland. Previous studies show that regions in Scotland differ for health-related traits, such as obesity, and display differences in lifespan of up to 7.5 years in men and 4.9 years in women, with both genetics and lifestyle differences being possible causal factors for regional health variation. Using data on obesity-related traits as indicators of the health status of ~11,000 Scottish individuals with genome-wide genetic information and records of possible causal lifestyle and socio-economic factors, the authors explored causes of regional variation. The authors combined genomic data with high fidelity and deep phenotypic, lifestyle and socio-economic data to address this question to demonstrate that both obesity traits and genetic variation differ significantly between regions. The authors note that for height, at least part of the variation between regions observed can be explained by genetics. However, regional variation for most obesity traits was not associated with regional genetic differences, but instead was associated with lifestyle and socio-economic variables, such as smoking, alcohol intake, diet and deprivation. The authors conclude that while genes and family environment are both important determinants of health-related traits, regional differences are attributable mostly to potentially modifiable environmental factors and that this may have important implications for healthcare policies, suggesting that inequalities can be tackled with appropriate social and economic policies and interventions. While this study is of potential interest to the readership of Nature Communications, there is no measure of study power and current study design would appear highly underpowered to exclude genetic contributions to the traits under study as some of the regional cohorts are very small, and since most of the life style, socioeconomic or geographic variables were actually not measured the conclusion that these variables explain the variation observed is preliminary and would require prospective validation. I have a few additional questions:

Questions/comments:

1. How many of the subjects analyzed were genotyped on Illumina Express array and how did these number vary from region to region - were other genotyping platforms used?
2. How many subjects were analyzed for each of the eight anthropometric vs three metabolic traits reported and how many geographic covariates and environmental covariates were analyze in subjects who were individually genotyped and how did these number vary from region to region
3. The authors report that for 9 of the 11 traits studied, differences between council areas were significant – what is the n for each of the council areas that ended up being analyzed.
4. The heritability analysis (shown in Table 1) would appear to support stronger genetic component (h^2) than environmental (as previously reported in many other studies), yet the authors conclusion is in conflict with the data.
5. How genetically homogeneous are these regional populations in Scotland and how did the authors rule out genomic inflation that may have adversely impacted their conclusions.

Reviewer #1

In their study, entitled “Regional variation in health is driven by lifestyle, not genetics”, Dr. Amador and colleagues examined differences in health-related traits and diseases between countries and between regions within countries, focusing on Scotland. Previous studies show that regions in Scotland differ for health-related traits, such as obesity, and display differences in lifespan of up to 7.5 years in men and 4.9 years in women, with both genetics and lifestyle differences being possible causal factors for regional health variation. Using data on obesity-related traits as indicators of the health status of ~11,000 Scottish individuals with genome-wide genetic information and records of possible causal lifestyle and socio-economic factors, the authors explored causes of regional variation. The authors combined genomic data with high fidelity and deep phenotypic, lifestyle and socio-economic data to address this question to demonstrate that both obesity traits and genetic variation differ significantly between regions. The authors note that for height, at least part of the variation between regions observed can be explained by genetics. However, regional variation for most obesity traits was not associated with regional genetic differences, but instead was associated with lifestyle and socio-economic variables, such as smoking, alcohol intake, diet and deprivation. The authors conclude that while genes and family environment are both important determinants of health-related traits, regional differences are attributable mostly to potentially modifiable environmental factors and that this may have important implications for healthcare policies, suggesting that inequalities can be tackled with appropriate social and economic policies and interventions.

We would like to thank the reviewer for the valuable comments. We understand that the reviewer considers the manuscript of interest for Nature Communications readers, but has some concerns. We tried to clarify all the points raised below and we also modified the text of the manuscript accordingly to make it easier to understand.

While this study is of potential interest to the readership of Nature Communications, there is no measure of study power and current study design would appear highly underpowered to exclude genetic contributions to the traits under study as some of the regional cohorts are very small, and since most of the life style, socioeconomic or geographic variables were actually not measured the conclusion that these variables explain the variation observed is preliminary and would require prospective validation.

The genetic contributions to the traits are studied at a population level. Genotypes for the 11,118 individuals were available and they were included in the study modelled as similarity matrices. In order to correct for possible stratification, the *geographical principal components* were included, although they did not explain much variation in the traits. From a genetic perspective, because we fitted the genetic matrices and principal components in the model, we obtained reasonable estimates of the heritability, which suggests that the genetic component is taken into account in an appropriate manner so the genetic contributions should be properly estimated irrespective of the size of the region because all individuals are included together in all the analyses.

Because some of the regions contain a smaller number of individuals, lack of power would imply inability to detect mean differences between regions but it would not be capable of creating spurious differences. Since we observe significant differences between the regions we conclude that we have enough power and the conclusions of the study are correct.

It is true that our environmental variables cover only part of the possible environments affecting individuals. However, they represent important known factors affecting obesity related traits and they explain the regional differences. The significance of these variables could be due to the variables themselves or some other that are correlated to them (we tried to clarify this on line 162).

We firstly ruled out the genetics being the major cause of the regional differences (line 59, line 139), and then we showed that the measured environmental variables can account for the differences. The fact that not all environmental variables are measured would be problematic if they weren't capable of explaining the regional differences (in that case we could not completely discard that the cause is environmental) but if the available variables can explain the differences, adding the non-measured ones could result in the model explaining more variance but would not alter the current conclusion that environmental factors can explain the majority of regional differences but genetics cannot. It could be possible that genetics had a small impact in the differences that we cannot detect but we are certain that the environmental factors are responsible for the main differences between regions.

I have a few additional questions:

Questions/comments:

1. How many of the subjects analyzed were genotyped on Illumina Express array and how did these number vary from region to region - were other genotyping platforms used?

9,863 samples used the Illumina HumanOmniExpressExome-8 v1.0 BeadChip and the remainder were genotyped using the Illumina HumanOmniExpressExome-8 v1.2 BeadChip. Both arrays were very similar, and only SNPs common to both were used (line 179). All 11,118 individuals included in the analyses had genotypes available. The distribution in regions is shown in Table S3.

2. How many subjects were analyzed for each of the eight anthropometric vs three metabolic traits reported and how many geographic covariates and environmental covariates were analyzed in subjects who were individually genotyped and how did these number vary from region to region

Thanks for the suggestion, this information is relevant and it is now included in the submission. There were some missing values for each trait (ranging between 38 and 436), with total number of individuals analysed being >10,600 for all traits. Supplemental Table 3 was modified to include this information. The numbers for each trait are not very different from the overall numbers shown originally in Table S3.

3. The authors report that for 9 of the 11 traits studied, differences between council areas were significant – what is the n for each of the council areas that ended up being analyzed.

This information is now shown in Supplemental Table 3 (see comment above).

4. The heritability analysis (shown in Table 1) would appear to support stronger genetic component (h^2) than environmental (as previously reported in many other studies), yet the authors conclusion is in conflict with the data.

As the reviewer noticed, our study shows that the heritabilities for all the traits ranged between 0.32 (for ABSI) and 0.87 (for height), with most of the values being near 0.50. The values are in concordance with previous studies for these traits suggesting that for them on average variation would be approximately 50% genetic and 50% environmental and that genetics is correctly taken into account. These moderate heritabilities do not necessarily have any implications for the cause of regional differences: since both genetics and environment contribute, approximately equally, to variation in the trait, both sources could be the cause of the regional differences (this statement remains valid as long as the genetic contributions are not 100%). That was the question we explored in the manuscript, modelling at the same time genetic and environmental components. We think an important conclusion from the study is that although both genes and environment make substantial contributions to variation of these traits within the population, for most traits it appears to be environmental factors that cause significant variation between regions. A second important observation is that studies that explore the causes of regional inequalities should account for both genetic and environmental factors if they are to avoid reaching biased conclusions (line 146).

We included in our analyses the genetic component (as two genetic similarity matrices) and a number of environmental variables. While many more environmental variables exist and are not available in our data, the genetic seem to be capturing all the genetic variance (based on previous h^2 estimates), so we are quite confident that our matrices represent quite well the genetic component in our population.

5. How genetically homogeneous are these regional populations in Scotland and how did the authors rule out genomic inflation that may have adversely impacted their conclusions.

The cohort is not substantially genetically heterogeneous as shown in a previous publication (Amador et al, 2015, BMC Genomics 16:437), although some stratification can be detected when using the appropriate markers and set of individuals. For that reason, we used the *Geographical Principal Components* firstly described in Amador et al. that represent and correct for the stratification in all the analyses. However, as shown in Amador et al. and in this manuscript, they do not explain much variance, confirming the homogeneity of the population. Given that the *Geographical Principal Components* and GRM are fitted in all the analyses, we can rule out false positives due to population stratification.

Reviewers' Comments:

Reviewer #1:

Remarks to the Author:

The authors have been responsive to the comments raised and have addressed and revised the manuscript accordingly and clarified the genetic aspects. The conclusions that environmental factors are the main drivers of the regional differences observed are in keeping with expectations as these regional populations are relatively homogeneous and emphasize the importance that these environmental variables are taken into account in studies of complex genetic disorders such as obesity, CVD and alike. I have no further comments

Reviewer #3:

Remarks to the Author:

I read both the paper Amador et al wrote and the review and responses that have already been elicited with interest. I think this is an interesting question to ask, and that the authors have a wonderful data set in order to conduct this analysis.

I do have, however, some questions about their method. It seems to me that the study in question is effective one of deducing confounding. Health status and region are correlated with each other, and this correlation might be explained by either a) Genetics, b) Environmental differences, c) Some combination of the two. Using the data in the analysis the authors settle on b) as the most likely.

Usually when considering a factor as a confounding factor it is not a case of whether the association remains significant after adjusting (though this would of course be the case for complete confounding), rather it is about the change of the effect estimate after adjustment.

As I understand it from the analysis that the authors have done they took these steps - 1) Tested a series of traits to see if postcode had an effect; 2) Added a genetics matrix to the model and checked what remained significant; 3) Added a number of relatedness matrices and checked what remained significant; 4) Added an environmental index based on post code and checked what remained significant. They do not (as far as I can see) have a model that only adjusts for the environment, and not the genetics - which would be useful to understand the effects.

I'm not sure the data is presented in a way that makes it possible for the reader to see the actual answers to these questions. Firstly, rather than looking at significance, the focus should be on adjustment. In this case the estimate would presumably be the proportion of the variance explained. Secondly I'm trying to work out if the environmental index, since it is based on postcode, is dimensionally different from postcode - i.e. would it not be better to look at the measured variables that authors think are important. This is especially the case as the SIMD which seems to contain a number of inputs that might be thought of as downstream of these anthropometric and health factors rather than upstream of it.

In short, the differences in regional phenotypes could be due to both genetics and environment, it doesn't require either to completely explain the differences (even though a model with both accounted for does appear to in most cases), and that by more fully exploring the adjustment in effects the authors are likely to get a better answer to the question they set out with.

Specific points:

Figure 1. The units of the standardized BMI aren't very informative. If the authors are going to use adjusted means then it would be better to calculate these to be actual BMI units.

Figure 2. Why were these four chosen? I think it would be much more interesting to look at the

example for height were there was a different pattern rather than four showing basically the same.

Line 191. A lot of these traits seem like they would be highly correlated with each other - why did the authors use so many? It would be useful to have the correlations of these in the study included in the supplement.

Line 208. The way that this scans at the moment suggests SIMD is a component of itself.

Lines 241-273. This is very confusing. The authors keep taking about 3 models, but these appear to be a different three models in 248-250 as they are from 265-273. In the first set all models have a genetic term, the latter seems to have at least one model with no genetic term. This is made all the more confusing by Table 1 (see more details below) where the caption talks about four models, but the table only have three columns of results, which again don't seem to completely overlap with the models described here.

Table 1. Firstly - why do the authors report significance measures as $-\log_{10}$ p-values? It is much more conventional to report p-values, and readers know what they are looking at. Incidentally, my quick calculations suggest that a $-\log_{10}$ value of 1.15 is not significant at the conventional <0.05 level, so the authors should indicate what threshold was used.

Secondly - there needs to be clarity about the models (see above) between the caption and the data in the table.

Thirdly - as I mentioned in the beginning of the review, the interest is not about whether something remains significant, but whether the predicted effect changes. By just reporting measures of significance this obscures the assessment of this. In fact the magnitude of the change in the level of significance leads me to suspect that there might be some important changes in effect estimate.

Extended Data Table 1

I would be useful if the column headers (G, K, C etc), or the caption gave a brief description of what each were, to make the figure more easy to understand as a standalone item.

Reviewer #3 (Remarks to the Author):

I read both the paper Amador et al wrote and the review and responses that have already been elicited with interest. I think this is an interesting question to ask, and that the authors have a wonderful data set in order to conduct this analysis.

I do have, however, some questions about their method. It seems to me that the study in question is effective one of deducing confounding. Health status and region are correlated with each other, and this correlation might been explained by either a) Genetics, b) Environmental differences, c) Some combination of the two. Using the data in the analysis the authors settle on b) as the most likely.

Usually when considering a factor as a confounding factor it is not a case of whether the association remains significant after adjusting (though this would of course be the case for complete confounding), rather it is about the change of the effect estimate after adjustment.

As I understand it from the analysis that the authors have done they took these steps - 1) Tested a series of traits to see if postcode had an effect; 2) Added a genetics matrix to the model and checked what remained significant; 3) Added a number of relatedness matrices and checked what remained significant; 4) Added an environmental index based on post code and checked what remained significant. They do not (as far as I can see) have a model that only adjusts for the environment, and not the genetics - which would be useful to understand the effects.

We would like to thank the reviewer for all their valuable comments. Overall it looks that the results were presented in an unclear way and that led to some misunderstanding of some of the analyses performed. We have removed some superfluous results and included some additional results and also attempted to improve the description of the methods and results in order to clarify what we did and how we reached our conclusions. We hope that the paper is clearer now. The main changes appear in blue in the manuscript. Some more detailed answers to specific comments can be found below.

The analyses performed were as follows (and now illustrated in Figure 1):

1) Test the traits to check if region had an effect (region is derived from the postcodes at which an individuals lived which are then grouped into the 32 Scottish council areas) (we term this the Basal Model).

2) We corrected for a genomic relationship matrix to remove the effects of genetic similarity and family structure (Family Model). Ignoring this strong family structure in our sample before testing the other confounders would potentially lead to biased estimates and tests of significance and hence inappropriate conclusions. We phrased this differently in this version of the manuscript in order to accommodate, as the reviewer requested, a comparison including the socio-economic and lifestyle (SELS) variables but not the regional genetic structure (the gPCs). We tested again the effect of region. A reduced effect of region would imply that the kinship is responsible for the regional differences in the traits.

3) We corrected for the genetic similarity (with the genomic relationship matrix) and also the geographical principal components (gPCs) representing the regional genetic structure of the sample (Structure Model). The gPCs capture the genetic differences between the regions as shown in Amador et al 2015. We again tested the effect of region. A reduced effect of region would imply that the regional genetic structure is responsible for the regional differences in the traits.

4) We jointly corrected for genetic similarity with the genomic relationship matrix and the covariates representing socio-economic and lifestyle effects (SELS) in the Environment Model. We again tested the effect of region. A reduced effect of region here would imply that the environmental differences between the individuals are responsible for the observable

differences in the traits. Note that for most of the traits the regional differences disappeared when fitting the SELS variables.

5) We finally corrected for the genomic relationship matrix, the SELS variables and the gPCs (Structure + Environment Model) and tested the effect of region in order to compare with the two previous models.

In the previous version of the manuscript, for two traits with remaining differences (creatinine and ABSI) we fitted another matrix created using the postcode, but as the reviewer pointed out, although different from the regions that we used to test the regional differences, there was a big overlap between the two, making the test redundant (testing for something that was almost explicitly corrected for in the previous step). We hence removed the analyses based on the AREA matrix (based on postcodes) from the current manuscript.

Finally, we repeated the analyses including a large set of matrices (Genomic relationship matrix, G; kinship matrix, K; couple matrix, C; and siblings matrix, S) and the results corroborate the conclusions drawn by the above models.

I'm not sure the data is presented in a way that makes it possible for the reader to see the actual answers to these questions. Firstly, rather than looking at significance, the focus should be on adjustment. In this case the estimate would presumably be the proportion of the variance explained. Secondly I'm trying to work out if the environmental index, since it is based on postcode, is dimensionally different from postcode - i.e. would it not be better to look at the measured variables that authors think are important. This is especially the case as the SIMD which seems to contain a number of inputs that might be thought of as downstream of these anthropometric and health factors rather than upstream of it.

We agree with the reviewer that there is information over and above the significance of effects that helps the reader interpret and understand the results and thus we have expanded and attempted to clarify the results we present.

In terms of variance explained, the changes in heritability in the different models are included in the paper. As shown in Figure 3, we explored the differences in the variance capture by the matrices describing the genetic and family environmental structure before and after including the sets of covariates (gPCs and SELS). The heritability estimates did not change greatly (and all changes were non-significant) in any of the models (robust estimates), although in many of the models the regional differences became non-significant.

We added two new tables (Table 3 and Supplementary Table 5) including information on variance explained by the geographical principal components (gPCs) and by the environmental variables (SELS). The gPCs explain little variance for most of the traits (with the exception of height, see discussion in the manuscript in line 195) whereas the SELS explain more variance. The genetic differences between the regions do not seem to be significant in most of the cases. However, a variety of environmental variables capture variance while they explain the regional differences.

We added these results to the manuscript and tried to clarify the meaning and the conclusions that can be drawn from them.

In short, the differences in regional phenotypes could be due to both genetics and environment, it doesn't require either to completely explain the differences (even though a model with both accounted for does appear to in most cases), and that by more fully exploring the adjustment in effects the authors are likely to get a better answer to the question they set out with.

As mentioned before, in order to clarify this we added a new model including the environmental covariates and not the regional genetic structure covariates (Table 1, fourth column: Structure Model). We also included two new tables (Table 3 and Supplementary Table 5) including the proportion of the variance that each of the fitted covariates explained for each of the traits analysed. In the Structure Model a genomic relationship matrix is also included. We consider that the results without including one will be biased because of the high enrichment of kinship in our cohort.

Specific points:

Figure 1. The units of the standardized BMI aren't very informative. If the authors are going to use adjusted means then it would be better to calculate these to be actual BMI units.

The plot was created to illustrate the differences between regions. The values were obtained using the residuals of the GREML analyses from the log converted BMI after removing the effects of the matrices and fixed effects. Unfortunately, we cannot convert the values meaningfully to actual BMI units but we think that the colour emphasise the differences between regions and the change in them.

Figure 2. Why were these four chosen? I think it would be much more interesting to look at the example for height were there was a different pattern rather than four showing basically the same.

We chose those four traits because for the four of the SELS covariates showed significant effects. We do not expect a change in the variance components after including covariates that did not show and effect, but changes might be observed in traits were the SELS explained the regional differences. What we showed is that even in those where the differences were explained by the SELS covariates, the estimates of the variance components did not change substantially or significantly . The case in height was very similar, see the plot below. However, since the four plots were very similar we followed the referees advise and kept just two traits which can be displayed in a larger size.

Line 191. A lot of these traits seem like they would be highly correlated with each other - why did the authors use so many? It would be useful to have the correlations of these in the study included in the supplement.

As the reviewer suggests the traits are correlated although the correlations were not generally extreme and the correlations are now included in Supplementary Table 1b. The different traits might reflect different genetic and environmental components of obesity and indeed they showed different results in our analyses.

Line 208. The way that this scans at the moment suggests SIMD is a component of itself.

We changed the sentence to: Socioeconomic covariates are Scottish index of multiple deprivation (SIMD, a deprivation ranking based on living area (now in line 268) and we added the information on what contributes to the index to Supplementary Table 2.

Lines 241-273. This is very confusing. The authors keep taking about 3 models, but these appear to be a different three models in 248-250 as they are from 265-273. In the first set all models have a genetic term, the latter seems to have at least one model with no genetic term. This is made all the more confusing by Table 1 (see more details below) where the caption talks about four models, but the table only have three columns of results, which again don't seem to completely overlap with the models described here.

Thanks for the suggestion. We renamed the models and tried to make a simpler description of them making easier to connect the results and the methodology section. We added a new figure (Figure 1) with some basics of the models to help the reading and we unified the names throughout the manuscript.

Table 1. Firstly - why do the authors report significance measures as $-\log_{10}$ p-values? It is much more conventional to report p-values, and readers know what they are looking at. Incidentally, my quick calculations suggest that a $-\log_{10}$ value of 1.15 is not significant at the conventional <0.05 level, so the authors should indicate what threshold was used.

We removed the $-\log_{10}$ transformation and include directly p values in the table. The star in the value of 1.15 was indeed a typo, thanks for pointing this out.

Secondly - there needs to be clarity about the models (see above) between the caption and the data in the table.

As mentioned above we rename all the models and try to make the manuscript easier to follow. The names and brief descriptions now appear in Figure 1 and are homogeneous through the sections.

Thirdly - as I mentioned in the beginning of the review, the interest is not about whether something remains significant, but whether the predicted effect changes. By just reporting measures of significance this obscures the assessment of this. In fact the magnitude of the change in the level of significance leads me to suspect that there might be some important changes in effect estimate.

As mentioned in a comment before, we explored the changes in the variance explained by the matrices in the different models together with the variance explained by the covariates. We did not see big differences in the heritabilities between the different models suggesting that the estimates are robust irrespective of which covariates we include in the models.

While the regional genetic structure of the sample (captured by the gPCs) rarely captured any variance, the environmental variables (SELS) captured between 2 and 35% of the variance in obesity related traits. Inclusion of SELs removed the significant effect of region for most traits, but without biasing the heritabilities.

We tried to clarify this on the manuscript. (See Table 2, Table 3 and Supplementary Table 5 and lines 174 and 212).

Extended Data Table 1

I would be useful if the column headers (G, K, C etc), or the caption gave a brief description of what each were, to make the figure more easy to understand as a standalone item.

We have now included the description in the caption of the figure (Now Table 2) for clarification.

Reviewers' Comments:

Reviewer #3:

Remarks to the Author:

I think that this manuscript is considerably improved, and the increase in clarity is very much appreciated. I particularly like the idea of Figure 1 - I think more papers should have something similar.

That said there are a few places where I had some residual issues following, and could be made more explicit.

Specifically -

Table 2. The sentence "The third column shows the variance captured by each matrix when fitted together", surely this should refer to the third to sixth columns.

The authors switch between percentage and proportion in the manuscript and should stick to one or the other, in particular in Table 3.

I while I understand what the authors mean when they say geographic PCs, it might be clearer to just use the phrase "regional genetics structure" for these types of analysis.

This is a minor quibble as it looks quite nice; but if, as I think they do, the information in Figure 3 is the same as that in Table 2, then having Figure 3 is slightly superfluous.

One general thing, since the manuscript is reporting p-values (or flags of significance in some cases) and quite a lot them, I think the authors need to be upfront about the level of significance. I'm ok with not requiring a full Bonferroni correction or FDR (as I mentioned in my previous review I think the data shown in table 2 is more important), but since these are flagged it seems sensible to tell the reader what the criteria is.

Other points -

In the supplementary note I think it would be good to give more information about years of education, the range of values looks a bit odd - is it asking about years of secondary education? post-16 education? or total years of education?

Now I understand the analyses performed, the discussion is a little light weight. The authors reiterate the discovery of familial associations for height and BIA, but do not consider why this might be (could it be height as a results of the high heritability explained by common SNPs?). I think the focus on the public health aspect of the paper is great, but would have liked some more consideration especially of the associations seen in BIA - which is quite interesting. The idea that there are socio-geographical disparities that result in changes to health status isn't particularly novel; one thing that might push the paper to a policy audience is a comment on the current level geographically targeting of health interventions for obesity - this is mentioned, but not discussed.

I also think the discussion could benefit from a starting paragraph that provides more of an overview of the results (at the moment the introduction to the method is quite "methods" focused). The results are there, but they are split over different paragraphs and as this is quite a complicated paper it would be good to anchor a reader to the main messages.

We respond in blue font to reviewer's #3 remarks, presented in black font.

Reviewer #3 (Remarks to the Author):

I think that this manuscript is considerably improved, and the increase in clarity is very much appreciated. I particularly like the idea of Figure 1 - I think more papers should have something similar

That said there are a few places where I had some residual issues following, and could be made more explicit. Specifically –

Table 2. The sentence "The third column shows the variance captured by each matrix when fitted together", surely this should refer to the third to sixth columns.

Thanks for pointing this out. It is now corrected in the caption.

The authors switch between percentage and proportion in the manuscript and should stick to one or the other, in particular in Table 3.

We changed the values in Table 3 and Supplementary Table 5 to proportion instead of percentage to match the others.

I while I understand what the authors mean when they say geographic PCs, it might be clearer to just use the phrase "regional genetics structure" for these types of analysis.

Done (lines 97 and 178).

This is a minor quibble as it looks quite nice; but if, as I think they do, the information in Figure 3 is the same as that in Table 2, then having Figure 3 is slightly superfluous.

We agree they have the similar information, but we think that the figure gives a quick idea of the results for a couple of traits and if the reader is interested they can go to the table for the complete set of values.

One general thing, since the manuscript is reporting p-values (or flags of significance in some cases) and quite a lot of them, I think the authors need to be upfront about the level of significance. I'm ok with not requiring a full Bonferroni correction or FDR (as I mentioned in my previous review I think the data shown in table 2 is more important), but since these are flagged it seems sensible to tell the reader what the criteria is.

We agree this is useful information. It is now included in the text (lines 83, 94, 104, 112).

Other points –

In the supplementary note I think it would be good to give more information about years of education, the range of values looks a bit odd - is it asking about years of secondary education? post-16 education? or total years of education?

A description of the Years of Education variable was added in the supplementary material (page 24).

Now I understand the analyses performed, the discussion is a little light weight. The authors reiterate the discovery of familial associations for height and BIA, but do not consider why this might be (could it be height as a results of the high heritability explained by common SNPs?). I think the focus on the public health aspect of the paper is great, but would have liked some more consideration especially of the associations seen in BIA - which is quite interesting. The idea that there are socio-geographical disparities that result in changes to health status isn't particularly novel; one thing that might push the paper to a policy audience is a comment on the current level geographically targeting of health interventions for obesity - this is mentioned, but not discussed.

We expanded the discussion with regard of height and particularly BIA Fat (line 207) and the geographically targeting of policies (line 235)

I also think the discussion could benefit from a starting paragraph that provides more of an overview of the results (at the moment the introduction to the method is quite "methods" focused). The results are there, but they are split over different paragraphs and as this is quite a complicated paper it would be good to anchor a reader to the main messages.

We added a new paragraph at the beginning of the discussion to wrap up the messages that we expand on later.